# Characteristics of In Vitro Culture and In Vivo Confocal Microscopy in Patients with Fungal Keratitis in a Tertiary Referral Hospital in Central China

**DOI:** 10.3390/microorganisms11020406

**Published:** 2023-02-06

**Authors:** Jia-Song Wang, Ya-Li Du, Nan Deng, Xi Peng, Hang Wong, Hua-Tao Xie, Ming-Chang Zhang

**Affiliations:** 1Department of Ophthalmology, Union Hospital, Tongji Medical College, Huazhong University of Science and Technology, Wuhan 430022, China; 2Department of Clinical Laboratory, Union Hospital, Tongji Medical College, Huazhong University of Science and Technology, Wuhan 430022, China

**Keywords:** Fungal Keratitis, In Vivo Confocal Microscopy, Fungal Culture, *Fusarium*, *Aspergillus*, *Alternaria*

## Abstract

Purpose: To investigate the characteristics of in vitro culture and in vivo confocal microscopy (IVCM) in patients with fungal keratitis (FK) presented in a tertiary referral hospital in central China. Methods: In this noncomparative retrospective study, patients with the diagnosis of FK between October 2021 and November 2022 were reviewed. An IVCM and fungal culture (corneal scraping specimens) were performed, and the characteristics were analyzed. Results: During October 2021 and November 2022, 85 patients were diagnosed with FK. From 63 culture-positive cases, 8 species of fungus were identified. The proportions of isolated fungal species were *Fusarium* and *Aspergillus* equally accounting for 33.3% (21 of 63), *Alternaria* 9.5% (6 of 63), *Curvularia* 6.3% (4 of 63), *Scedosporium apiospermum* 6.3% (4 of 63), *Paecilomyces lilacinus* 3.2% (2 of 63), *Exserohilum* 3.2% (2 of 63), and *Candida* 4.8% (3 of 63), respectively. In positive culture cases, IVCM was found to be positive for hyphae or spores in 61 of 63 patients (96.8%). Different fungal species had a variety of cultural characteristics and IVCM manifestations. Conclusions: In a tertiary referral hospital in central China, *Fusarium* species, *Aspergillus* species, and *Alternaria* species were the 3 most common isolated fungal pathogens, and the proportion of *Aspergillus* species was significantly higher than that in other regions of China. Careful lesion depth examination by IVCM and OCT should be taken before lamellar keratoplasty to avoid postoperative recurrence. Identifying the IVCM image and culture characteristics will facilitate rapid diagnosis and proper treatment, but IVCM cannot yet replace fungal cultures to distinguish between different fungal species.

## 1. Introduction

Fungal keratitis (FK) is a severe corneal infection that often results in blindness and eye loss in developing countries [1,2]. Annually, more than 100,000 eyeballs are removed due to various factors associated with FK, including delayed fungal identification, a multitude of microbiological species, poor efficacy and permeability of antifungal drugs, and drug-resistance [1,3,4]. Those most susceptible to this condition are typically agricultural and outdoor workers in tropical or subtropical regions [2].

Though time-consuming, microbiological culture of corneal scrapings is still the gold standard for identifying the causative organism in FK [5]. In vivo confocal microscopy (IVCM) could help ophthalmologists diagnose FK immediately, especially for filamentous fungi [6]. Over 100 different species could cause FK in humans [7]. Pathogens show different patterns according to geographical location and climatic conditions [8,9,10,11], and the fungal spectrum is changing [12,13].

In order to improve the prognosis of FK, ophthalmologists are required to gain a better understanding of the characteristics of the isolation and identification of fungi, as well as those of IVCM [6,7,14]. However, related data about fungal pathogens have been limited in central China in the last decade. Therefore, this study aims to describe the distribution of pathogen species and IVCM characteristics of FK in a tertiary referral hospital in central China.

## 2. Materials and Methods

### 2.1. Participants

The medical history data of patients with the diagnosis of FK at Union Hospital, Tongji Medical College, Huazhong University of Science and Technology between October 2021 and November 2022 were retrospectively reviewed. The Ethics Committee of Union Hospital approved this study (Approval No. UHCT22928) according to the tenets of the Declaration of Helsinki.

All patients underwent anterior segment photography, IVCM, corneal scraping, anterior segment optical coherence tomography (AS-OCT), and bacterial and fungal culture examinations. The inclusion criteria were: (1) hyphae or spores observed on IVCM; (2) hyphae or spores detected by corneal scraping; or (3) a positive fungal culture result. The exclusion criteria were the following: (1) other types of keratitis; or (2) fungal endophthalmitis without keratitis. The culture-positive cases (corneal scraping specimens) were collected for further analysis.

The following clinical data were collected: (1) demographic data (age, sex, risk factors, time of onset, history of treatment, previous history of eye disease and surgery); (2) ocular examination data (area of epithelial defect, width and depth of infiltration, presence and height of hypopyon, and presence of endothelial plaque and satellite lesion); (3) treatment processes; (4) visual changes on follow-up.

### 2.2. In Vivo Confocal Microscopy

An IVCM was performed to detect fungal hyphae and spores as previously described [15], using the Heidelberg Retina Tomograph II with Rostock Cornea Module (Heidelberg, Germany). All the operations were performed by Dr. J.-S. W. Since dense lesions might affect the observation, the lesions along the edge of the ulcer were scanned to improve the positive diagnostic rate. After topical anesthesia, the depth and x–y position of scanning were manually controlled. For filamentous fungi, a positive result was defined as observation of branching linear structures with varying reflectivity and length. For yeast, a positive finding was identified as detection of numerous, high-contrast elongated particles with pseudohyphae. For each participant, all measurable hyphae or pseudohyphae present in all section images were collected for subsequent analysis. The length and width of hyphae or pseudohyphae and branch angle were calculated by Image J software (National Institutes of Health, Bethesda, MD, USA).

### 2.3. Microbiological Culture and Diagnosis

Following a standard microbiological procedure [16], a corneal microbiological test under sterile conditions was performed by scraping the ulcer’s edges to obtain the tissue (J.-S. W.) and inoculating the tissue onto Sabouraud dextrose agar or blood agar for culture. Then, the species was distinguished by microscopy and lactophenol cotton blue staining. Fungal culture positivity was reported by Dr. N. D. if any of the following criteria were met: (a) growth of the same fungal species on ≥1 solid media or (b) semiconfluent growth at the site of inoculation in 1 solid medium.

### 2.4. Medical Treatment and Surgical Indications

Topical eyedrops were administered to patients diagnosed with FK, including 0.15% amphotericin B (XinYa, Shanghai, China) and 1% Voriconazole (Pfizer, New York, NY, USA) hourly for 2 weeks, every 2 hours for 4–8 weeks. Oral Voriconazole was administered at 200 mg twice daily for 4 weeks. The response to therapy was noted on slit lamp examination. To avoid posterior synechiae of the iris and reduce inflammation, 1% atropine and 0.1% bromfenac sodium eye drops were used. If there were no changes or worsening of corneal lesions after 7 days of medical treatment, the operation would be performed.

### 2.5. Postoperative Management and Follow-Up

All patients in this study were examined and followed with hepatic and renal function regularly monitored. All complications were documented and described. The FK healing was defined as the re-epithelialization, disappearance of infiltrate and hypopyon, absence of hyphae through IVCM. Failure was defined as corneal perforation or recurrence of FK or eyeball removal. The outcome measures included visual acuity, healing of the ulcers, recurrence of FK, and relevant complications. All patients were followed up daily for 1 week, biweekly for 1 month, monthly for 3 months, and then at different intervals after treatment.

### 2.6. Statistical Analysis

All aggregated data were expressed as mean ± standard deviation (SD). This was an observation descriptive analysis. No statistical analysis was performed.

## 3. Results

### 3.1. Demographics and Predisposing Factors

During October 2021 and November 2022, 85 patients were diagnosed with FK. The average age of the patients was 60.1 ± 7.8 (30–79) years. The number of female patients (37, 43.5%) was lower than that of male patients (48, 56.5%). All the patients had unilateral infections (44 right eyes, 51.8%). Patients from rural areas accounted for 88.2% of the total (75 of 85). In 56.5% (48 of 85) of the cases, plant-related trauma was the leading risk factor, and corn leaves were the primary source of trauma in 21.1% (18 of 85) patients. In 2 patients (2.4%), it was due to long-term use of topical steroids. No significant predisposing factors were recorded in 17 patients (20%) (Table 1).

### 3.2. Microbiological Culture Findings

All suspicious FK patients who visited the hospital underwent fungal culture, and the positive rate was 74.1% (63 of 85). Among 63 culture-positive cases, 8 species of fungus were identified. *Fusarium* (Figure 1A) and *Aspergillus* (Figure 1B) were the main cultivated species, both accounting for 33.3% (21 of 63).

The proportions of other fungal species were *Alternaria*, 9.5% (6 of 63) (Figure 1C), *Curvularia*, 6.3% (4 of 63) (Figure 1D), *Scedosporium apiospermum*, 6.3% (4 of 63) (Figure 2A), *Paecilomyces lilacinus*, 3.2% (2 of 63) (Figure 2B), *Exserohilum*, 3.2% (2 of 63) (Figure 2C), and *Candida*, 4.8% (3 of 63) (Figure 2D), respectively. The data of isolated fungal species are shown in Table 2.

*Fusarium* species were typically flat, spreading, wooly colony growths, initially appearing white or grey-white in color, then producing a variety of colors when grown in culture (Figure 1A). *Fusarium* directly produced long chains with large numbers of conidia together with mycelia (Figure 1A) in a process known as adventitious sporulation [17]. *Aspergillus* branched directly, called dichotomous branches [18], instead of producing only lateral branches and grew gray-green to dark-green in color with white border, hairy colonies and conidia located on the branched conidiophores (Figure 1B). *Alternaria* species colony on Sabouraud dextrose initially appeared as gray flocculated mold before first changing to green-brown then to dark-brown, with a loose surface and gray-white border, lactophenol blue staining showing long chains of abundant rod-like conidia with multiple transverse, longitudinal, and oblique spacers (Figure 1C). Many conidia had a pointed proboscis at the distal end, and the conidiophores were primarily unbranched, and no secondary conidiophores were present (Figure 1C). Sabouraud agar plate culture showed the *Curvularia* colonies initially presented white flocculent aerial hyphae and changed to yellowish-brown or black after ripening (Figure 1D). Conidiophores were often branched, uniformly brown, septated and curved (Figure 1D).

Fungal colonies of *Scedosporium apiospermum* were spreading, wooly colony growths with white at the edges and olive-green umbrella-like at the center (Figure 2A). Microscopic appearance with lactophenol cotton blue stain showed septated hyphae with elongated meridiangium, and single conidia (Figure 2A). Conidia were elliptic and single-celled, with a large end apical (Figure 2A). Blood agar plates of *Paecilomyces lilacinus* showed the colonies presented a white to lilac in color and a floccose aerial mycelium (Figure 2B). Long chains, small round to fusiform conidia were shown in lactophenol cotton blue staining (Figure 2B). The colony morphology of *Exserohilum species* on Sabouraud agar plate was white in the center and brown surrounding feather appearances, and microscopy revealed numerous decolorized fungi with lack of color and *Exserohilum rostratum* (Figure 2C). When grown on Sabouraud dextrose agar and blood agar plates, *Candida* appeared much as it does in corneal tissue: smooth, glossy, raised, cream-colored colonies clustered together at close range, microscopy showing a purple fusiform appearance with pseudohyphae (Figure 2D).

### 3.3. In Vivo Confocal Microscopic Characteristics

In all patients with positive cultures, IVCM was found to be positive for hyphae or spores in 61 patients, and the positivity rate was 96.8% (61 of 63). Different fungal species had a variety of IVCM manifestations. Some reports have postulated that fungi’s hyphae length, width, and branching, as seen in IVCM images, might be used to differentiate fungal species [6,19].

Thirty clear IVCM images were collected and measured for each fungal species. The IVCM images obtained from culture-positive *Fusarium* keratitis revealed a large number of highly reflective lines that were 374.3 ± 49.2 (*n* = 30, ranging from 320 to 500) μm in length, and 9.4 ± 2.3 (*n* = 30, ranging from 5 to 12) μm in width, branching at 45.4 ± 7.8 (*n* = 30, ranging from 30 to 56) degrees (Figure 3A). The IVCM examination of *Aspergillus* keratitis showed numerous high-contrast lines that were 262.8 ± 20.6 (*n* = 30, ranging from 220 to 297) μm in length, and 11.7 ± 2.7 (*n* = 30, ranging from 6 to 15) μm in width, with branch angle at 52.1 ± 5.5 (*n* = 30, ranging from 42 to 59) degrees (Figure 3B). The IVCM examination of *Alternaria* keratitis showed many hyperreflective filaments, and the filaments were 7.9 ± 1.8 (*n* = 30, ranging from 5 to 10) μm in width, and 130.7 ± 45.7 (*n* = 30, ranging from 68 to 188) μm in length with branch angle of 39.5 ± 3.7 (*n* = 30, ranging from 33 to 46) degrees (Figure 3C). The IVCM of *Curvularia* keratitis showed short and thick hyphae that were 12.5 ± 1.8 μm (*n* = 30, ranging from 8 to 15) μm in width, and 37.3 ± 8.6 (*n* = 30, ranging from 20 to 48) μm in length without apparent fungal branching (Figure 3D).

The IVCM of the *Pseudallescheria boydii* keratitis showed hyper-reflective, branching, and interlocking structures and typical fungal hyphae that were 7.8 ± 1.7 (*n* = 30, ranging from 5 to 10) μm in width, and 217.3 ± 56.5 (*n* = 30, ranging from 126 to 292) μm in length with a branch angle of 48.2 ± 6.0 (*n* = 30, ranging from 35 to 57) degrees (Figure 3E). The IVCM of *Paecilomyces lilacinus* keratitis showed linear structures with dichotomous branching fungal hyphae that were 6.3 ± 2.2 (*n* = 30, ranging from 3 to 10) μm in width and 71.2 ± 22.9 (*n* = 30, ranging from 25 to 95) μm in length with a branch angle of 63.7 ± 17.2 (*n* = 30, ranging from 29 to 88) degrees (Figure 3F). The IVCM of *Exserohilum* keratitis showed numerous large septate hyphae that were 8.0 ± 1.6 μm (*n* = 30, ranging from 6 to 10 μm) in width and 320.2 ± 50.6 (*n* = 30, ranging from 210 to 380) μm in length with a branch angle of 82.1 ± 7.2 (*n* = 30, ranging from 66 to 90) degrees (Figure 3G). The IVCM examination of *Candida* keratitis showed numerous elongated particles, coral-like, without branching (Figure 3H).

### 3.4. Treatment Methods and Visual Outcomes

A total of 20.6% (13 of 63) of the patients improved after topical drug therapy. The rest, 79.4% of patients (50 of 63), with deep or wide lesions responded poorly to medical treatment, and surgical intervention was performed (Table 3).

Ten patients underwent corneal debridement and intrastromal injection of voriconazole. Another 26 patients underwent conjunctival flap covering surgery after receiving corneal debridement and intrastromal injection of voriconazole. Twelve patients received deep anterior lamellar keratoplasty (DALK). Among these patients, 2 underwent eyeball removal, and both had positive fungal culture results for *Pseudallescheria boydii* (Table 3).

Visual acuity change is an essential indicator of improvement in patients with FK after treatment. A total of 35 of 63 (55.6%) patients had improved visual acuity after treatment, 26 of 63 (41.2%) patients had no change in visual acuity, and 2 of 63 (3.2%) patients had decreased visual acuity after treatment. The lack of improvement was often due to pupillary lesions or other serious complications (Table 3).

## 4. Discussion

The most common cause of FK is an exogenous source in origin, usually due to fungal fragments or spores implanted into the cornea’s surface layer. Patients with FK are generally healthy, young male agricultural or outdoor workers who accidentally suffer from plant-related trauma during harvesting [1]. The proportion of patients from rural areas was 88.2% in our study. Males (56.5%) were predominant in the current study because they make up a more significant proportion of agricultural and outdoor workers than females, which is consistent with other studies [11,20,21,22,23,24]. Among other reported preconditions for FK, using contact lenses is a significant risk factor in developed countries [1,13,14,22,25,26,27].

The common pathogenic species included *Aspergillus*, *Fusarium*, *Candida*, *Curvularia,* and *Penicillium*, with *Fusarium* (37–62%) and *Aspergillus* (24–30%) being the most common ones [11,28,29]. *Aspergillus*, *Fusarium*, and *Curvularia* species are found mainly in tropical regions, while yeast is common in temperate areas of the world [30]. In developed countries, the prevalence of Candida keratitis was reported to be 60.6% [21]. Prajna et al. reported that *Aspergillus* species were the most common etiologies in northern and eastern India, while the *Fusarium* species were the most ordinary pathogens in western and southern India [31]. *Fusarium* and *Aspergillus* species were 2 of the most common isolated pathogens in our study, followed by *Alternaria* species, consistent with a previous report from central China [9].

Fungal species are influenced by differences in geography, climate, age, sex, socioeconomic status, agricultural activity, and degree of urbanization, so there are significant regional variations. In our research, Hubei, a major agricultural province in central China, has a subtropical climate and an extraordinarily higher positive rate of *Aspergillus* species than other regions of China. The proportion of *Aspergillus* species ranges from 7.3% [29] to 21.7% [10] in northern China, while this figure was 15.2% [32]) to 30.7% [10] in southern China. This indicates that the proportion of *Aspergillus* species increases with regional latitude.

In all cases of clinically suspected fungal infection, we found clear evidence of fungal hyphae and spores on IVCM in 61 patients, which significantly compensates for the shortcomings of fungal culture. The positivity rate of IVCM was 96.8% in this study. The IVCM is a promising complementary diagnostic approach of increasing importance that allows non-invasive real-time direct visualization of potential fungal pathogens and shows deeper layers directly in the cornea. Comparing culture plates in vitro and IVCM images in vivo may be a method to determine the morphological features of different species, but to date, such information is lacking in the published literature [6]. The IVCM can distinguish between filamentous and non-filamentous fungal species. In the future, more studies with IVCM images of specific fungal species may facilitate the identification of features specific to different species. Thus, fungal culture remains the gold standard for identifying pathogenic organisms [1,21].

After topical drug treatment, 20.6% of patients showed improvement, and the most frequent initial therapy was 0.15% amphotericin B and 1% Voriconazole eye drops. Natamycin is the only drug approved by the FDA for the treatment of corneal infection caused by fungi. However, it has difficulty penetrating the corneal stroma and is not available in some areas. Thus, other antifungal drugs, such as amphotericin B and fluconazole, are recommended as complementary treatments [33]. In our study, 79.4% of the patients received an operation, perhaps because the condition of patients in our hospital was more severe and the proportion of *Aspergillus* species was higher. Our results were similar to other reports that suggested corneal debridement combined with intrastromal voriconazole is a safe and effective treatment for FK [34,35]. Moreover, conjunctival flap surgery [36,37] and lamellar keratoplasty [8,38,39] are safe and effective in managing FK. Xie et al. reported *Fusarium* species are the most common isolated pathogens in northern China, and that *Fusarium* hyphae grow in the horizontal direction and *Aspergillus* hyphae show vertical growth [40,41]. Careful lesion depth examination by IVCM and OCT should be taken before lamellar keratoplasty to avoid postoperative recurrence.

The variety of fungal species in our study was not abundant. Fungal species in our region may exhibit characteristics that differ from other parts of the world. Finally, the small number of each fungus in this study makes it difficult to draw conclusions about the best treatment for the infections.

## 5. Conclusions

In a tertiary referral hospital in central China, *Fusarium*, *Aspergillus*, and *Alternaria* species were the 3 most common cultured fungal pathogens, and the proportion of *Aspergillus* species was significantly higher than that in other regions of China. Careful lesion depth examination by IVCM and OCT should be taken before lamellar keratoplasty to avoid postoperative recurrence. Identifying the IVCM and fungal culture characteristics will facilitate rapid diagnosis and proper treatment, but IVCM cannot yet replace fungal culture to distinguish different fungal species.

## Figures and Tables

**Figure 1 microorganisms-11-00406-f001:**
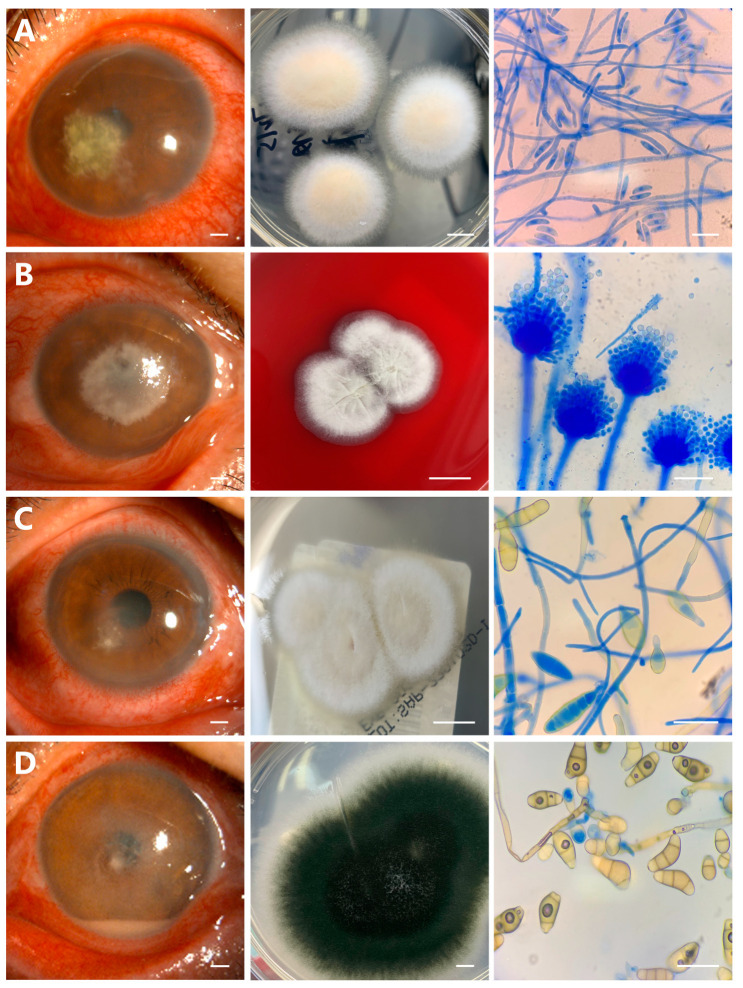
Anterior segment photography, colonies morphology, and lactophenol blue staining of microscopic characteristics of fungal culture species. (**A**) *Fusarium* species are flat, spreading, wooly colonies, and microscopic examination shows long chains with numbers of conidia and mycelia. (**B**) *Aspergillus* species grow gray-green to dark-green in color with a white border, hairy colonies, and microscopic examination shows conidia located on the branches. (**C**) *Alternaria* species colonies are gray flocculated mold, microscopic examination shows long chains of abundant rod-like conidia. (**D**) *Curvularia* species colonies are black and fluffy aerial hyphae, and microscopic examination shows conidiophores are branched, uniformly brown, septated and curved. (Scale bars = 1 mm in the left; 1 cm in the middle; 50 μm in the right column).

**Figure 2 microorganisms-11-00406-f002:**
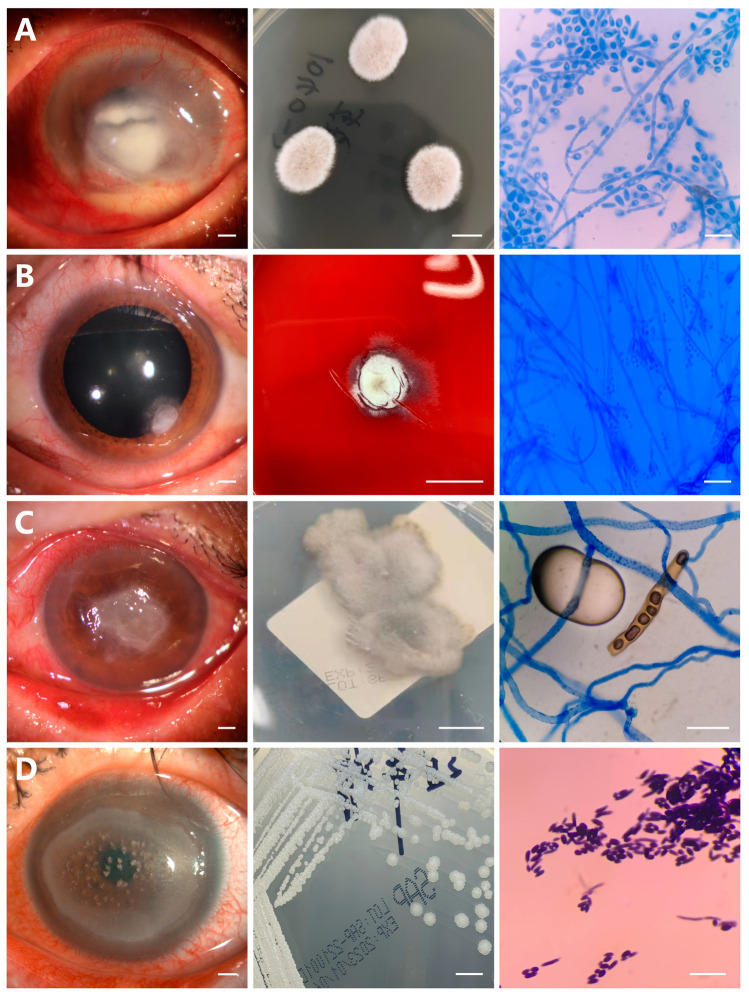
Anterior segment photography, colonies morphology, and lactophenol blue staining of microscopic characteristics of fungal culture species. (**A**) *Scedosporium apiospermum* colonies are white at the edges and olive-green umbrella-like at the center, flat, spreading, wooly colonies, and microscopic examination shows hyphae with elongated meridiangium and large end apical. (**B**) *Paecilomyces lilacinus* colonies are white to lilac in color and a floccose aerial mycelium, and microscopic examination shows long chains with small round to fusiform conidia. (**C**) *Exserohilum* species colonies are white in the center and brown surrounding feather appearances, microscopic examination shows numerous decolorized fungi with lack of color and *Exserohilum rostratum*. (**D**) *Candida* species are smooth, glossy, raised, cream-colored colonies and microscopic examination shows a purple fusiform appearance with pseudohyphae. (Scale bars = 1 mm in the left; 1 cm in the middle; 50 μm in the right column).

**Figure 3 microorganisms-11-00406-f003:**
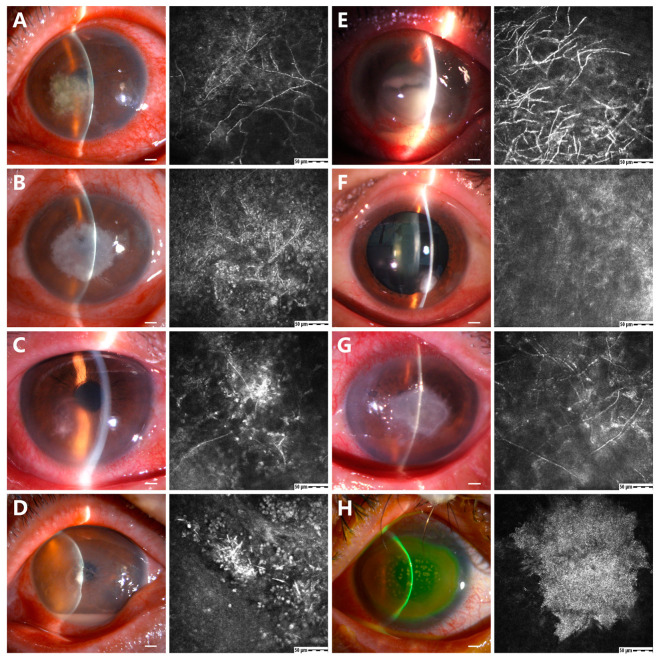
In vivo confocal microscopy (IVCM) images of patients with FK. (**A**) *Fusarium* shows a large number of highly reflective lines, 374.3 ± 49.2 μm in length, 9.4 ± 2.3 μm in width, branching at 45.4 ± 7.8 degrees. (**B**) *Aspergillus* shows numerous high-contrast lines, 262.8 ± 20.6 μm in length and 11.7 ± 2.7 μm in width, with branches at 52.1 ± 5.5 degrees. (**C**) *Alternaria* shows many hyperreflective hyphae, 7.9 ± 1.8 μm in width and 130.7 ± 45.7 μm in length, with branching at 39.5 ± 3.7 degrees. (**D**) *Curvularia* shows short and thick hyphae structures, 12.5 ± 1.8 μm in width and 37.3 ± 8.6 μm in length without apparent fungal branching. (**E**) *Pseudallescheria boydii* shows hyper-reflective, branching hyphae, 7.8 ± 1.7 μm in width and 217.3 ± 56.5 μm in length, with the angle of fungal branching at 48.2 ± 6.0 degrees. (**F**) *Paecilomyces lilacinus* shows linear structures, dichotomous branching hyphae, 6.3 ± 2.2 μm in width and 71.2 ± 22.9 μm in length, with the angle of fungal branching at 63.7 ± 17.2 degrees. (**G**) *Exserohilum* shows numerous large septate hyphae, 8.0 ± 1.6 μm in width and 320.2 ± 50.6 μm in length, with the angle of fungal branching at 82.1 ± 7.2 degrees. (**H**) *Candida* shows numerous elongated particles, coral-like, and without branching. (Scale bars = 1 mm in slit lamp; 50 μm in confocal microscopic pictures).

**Table 1 microorganisms-11-00406-t001:** Characteristics of patients with FK admitted to Wuhan Union Hospital from October 2021 to November 2022 (*n* = 85).

Characteristic	
Sex, *n*	
Male	48 (56.5%)
Female	37 (43.5%)
Age, y	
Mean ± SD	60.1 ± 7.8
Range	30–79
Eye, *n*	
Right	44 (51.8%)
Left	41 (48.2%)
Region, *n*	
Countryside	75 (88.2%)
City	10 (11.8%)
Hubei Province	77(90.5%)
Henan Province	5(5.9%)
Hunan Province	2(2.4%)
Jiangxi Province	1(1.2%)
Risk factor, *n*	
Plant/agriculture trauma	48 (56.5%)
Trauma from other items	18(21.1%)
Topical steroid	2 (2.4%)
Unknown	17 (20%)

**Table 2 microorganisms-11-00406-t002:** Fungal species isolated from culture-positive patients with FK (*n* = 63).

Microbiology, *n*	63
* Fusarium species*	21 (33.3%)
* Aspergillus species*	21 (33.3%)
* Alternaria species*	6 (9.5%)
* Curvularia*	4 (6.3%)
* Scedosporium apiospermum*	4 (6.3%)
* Paecilomyces lilacinus*	2 (3.2%)
* Exserohilum*	2 (3.2%)
* Candida parapsilosis*	3 (4.8%)

**Table 3 microorganisms-11-00406-t003:** Analysis of treatment modalities and changes in visual acuity in culture-positive patients (*n* = 63).

Treatment	
Medications	13 (20.6%)
Surgery	50 (79.4%)
Intrastromal injection + debridement	10 (15.9%)
Intrastromal injection + debridement + conjunctival flap	26 (41.3%)
Deep anterior lamellar keratoplasty	12 (19.0%)
Enucleation	2 (3.2%)
Change in visual acuity	
Improved	35 (55.6%)
Unchanged	26 (41.2%)
Decreased	2 (3.2%)

## Data Availability

The original contributions presented in the study are included in the article. Further inquiries can be directed to the corresponding author.

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
