# Peer review of "Characteristics of In Vitro Culture and In Vivo Confocal Microscopy in Patients with Fungal Keratitis in a Tertiary Referral Hospital in Central China"

_microorganisms, 2023, doi:10.3390/microorganisms11020406_

Round 1

Reviewer 1 Report

The article is well organized and I suggest a minor reversion before the publication.

1 The title“...in Central China, however, the patients seem to be from Hubei province, it seems more accurate to change “central China” into “Hubei province”. If not only from Hubei, it is recommended to add up detailed information in the "region" in Table 1.

2 Page 2, 2.1. Participants: The authors described the inclusion criteria in detail. Were there any exclusion criteria?

3 In order to reduce the experimental error, were the "IVCM" and "Microbiological Culture and Diagnosis" conducted by a special person?

4 Page 3: “2.5. Postoperative Management and Follow-Up. The author should tell us how to "follow-up"? For example, how long is the duration of the patient's return visit, and the observation period?

5 Page 3: 2.6. Statistical Analysis. The author should describe in more detail, such as t-test and analysis of variance.

6 What are the limitation of the article or the direction of the next efforts? Suggest that the authors state this in the discussion.

Reviewer 2 Report

The thought behind this article on fungal keratitis is very good, because it addresses a sight-threatening eye infection of increasing global incidence. However, the style and presentation should be improved to render it more comprehensible to your readers. The authors should ensure that all abbreviations used in the manuscript for the first time are written in full. Sentences that have grammatical/punctuation errors should be reworded. The authors should ensure that cited references relate to and support the claims made. I would like the authors to address these suggestions/comments raised by the reviewer.

Specific suggestions/comments raised by the reviewer:

Lines 46 – 47: The authors should reword this sentence to render it more comprehensible.

Lines 54 - 55: The authors should reword this sentence to render it more comprehensible.

Lines 74 – 75: Render “presence, and height of hypopyon” as “presence and height of hypopyon”

Lines 87 – 88: The authors should reword this sentence to render it more comprehensible.

Lines 94 – 97: Consider revising this run-on sentence to render it comprehensible.

Line 102: Provide an alternate rendering for “FK accepted topical eyedrops”

Lines 105 – 106: The authors should reword this sentence to render it more comprehensible.

Lines 125 - 126: The authors should reword this sentence to render it more comprehensible.

Lines 139 – 140: Provide an alternate rendering for this sentence.

Line 151: The authors should check that this cited reference relates to and supports the claim made, as the cited reference seems misplaced. Elaborate on this process of ectogenesis.

Lines 198 – 200: Re-evaluate the veracity of this statement.

Lines 249 – 250: Ten patients were given intrastromal injection of voriconazole, right?

Line 250: Elaborate on “Again on a basis”

Lines 274 – 276: The authors should reword this sentence to render it more comprehensible.

Lines 279 – 282: According to previous reports implies the need to cite more than one reference. Cite the other references for these reports.

Lines 292 – 294: Provide an alternate rendering for this sentence.

Figures 1 and 2: Looking at the colony pigmentation for figure 1A, it is not entirely white. There is a mix of other colors, and as such, provide a more accurate representation of the fungal colony pigmentation on the Sabouraud dextrose agar. For example, Aspergillus fumigatus usually appear as gray-green to dark-green in colour with a white border on the SDA. This accurate representation of the color of the fungal colony on SDA should apply to the rest of the fungal isolates in this study.

Round 2

Reviewer 2 Report

The authors have completed a good revision of the original manuscript. Sentences that have grammatical/punctuation errors should be reworded.

Specific suggestions/comments raised by the reviewer:

Lines 94 – 96: Consider revising this run-on sentence. Consider rendering it as “Following a standard microbiological procedure [16], a corneal microbiological test under sterile conditions was performed by scraping the ulcer's edges to get the tissue (J.-S. W.) and inoculating the tissue onto Sabouraud dextrose agar or blood agar for culture. Then, the species would be distinguished by microscopy and lactophenol cotton blue staining.

Lines 208 – 209: The authors she should reword this sentence to render it more comprehensible.

Line 210: Render “microscopic revealed” as microscopy revealed”
